# Pre-Coking Strategy Strengthening Stability Performance of Supported Nickel Catalysts in Chloronitrobenzene Hydrogenation

**Ping Wang [1], Shiyi Wang [1], Ronghe Lin [1], Xiaoling Mou [1,\*] and Yunjie Ding [1,2,3,\*]**

[1] Hangzhou Institute of Advanced Studies, Zhejiang Normal University, 1108 Gengwen Road, Hangzhou 311231, China; pingwang@zjnu.edu.cn (P.W.); wangshiyi@dicp.ac.cn (S.W.); catalysis.lin@zjnu.edu.cn (R.L.)

[2] Dalian National Laboratory for Clean Energy, Dalian Institute of Chemical Physics, Chinese Academy of Sciences, 457 Zhongshan Road, Dalian 116023, China

[3] The State Key Laboratory of Catalysis, Dalian Institute of Chemical Physics, Chinese Academy of Sciences, 457 Zhongshan Road, Dalian 116023, China

\* Correspondence: xiaoling.mou@zjnu.edu.cn (X.M.); dyj@dicp.ac.cn (Y.D.); Tel.: +86-411-84379143 (Y.D.)

**Abstract:** Supported nickel catalysts represent a class of important catalytic materials in selective hydrogenations, but applications are frequently limited by metal agglomeration or active-site blocking induced by the presence of hydrogen halides. Herein, we report a novel pre-coking strategy, exposing the nickel nanoparticles under methane dry reforming conditions to manipulate performance in the continuous-flow hydrogenation of 1,2-dichloro-4-nitrobenzene. Compared with the pristine nickel catalyst, the nanotube-like coke-modified nickel catalyst showed weakened hydrogenating ability, but much improved stability and slightly better selectivity to the target product, 3,4-dichloroaniline. Characterization results revealed that the strengthened stability performance can be mainly linked to the reduced propensity to retain chlorine species, which seems to block the access of the substrate molecules to the active sites, and thus is a major cause of catalyst deactivation on the pristine nickel catalyst. Coke deposition can occur on the pre-coked nickel catalyst but not on the pristine analog; however, the impact on the stability performance is much milder compared with that on chlorine uptake. In addition, the presence of coke is also beneficial in restraining the growth of the nickel nanoparticles. Generally, the developed method might provide an alternative perspective on the design of novel transition-metal-based catalytic materials for other hydrogenation applications under harsh conditions.

**Keywords:** pre-coking; nickel; hydrogenation; deactivation mechanism; chlorine accumulation

## 1. Introduction

Halogenated anilines and their derivatives, such as chloroanilines and dichloroanilines, represent a class of important high-value intermediates for the synthesis of pharmaceutical ingredients, synthetic rubber, dyes, pesticides, etc. [1–3] Typical synthetic routes for these highly functionalized molecules are based on a two-step approach, i.e., the nitration of chlorobenzenes or dichlorobenzenes followed by the chemoselective hydrogenation of the nitro group. The hydrogenation process is traditionally realized by the use of stoichiometric reducing agents, including Fe, Zn, Sn, or hydride reagents [4], which unfortunately is environmentally unsustainable due to the generation of substantial amounts of harmful chemical wastes. For this reason, the development of replacing heterogeneous catalysts is necessary to overcome this drawback. In this context, different catalytic systems have been reported, including precious metals (Pt [5–10], Pd [11–13], Ru [14,15], Ir [16], Au [17]), non-precious metals (Co [18–20], Ni [21,22], Fe [22–24]), and metal-free catalysts [25,26]. Among these, Pt-based precious metal catalysts have been widely studied, mainly owing to their superb hydrogenating ability and extremely high chemoselectivity, the latter representing one

of the main challenges in this field [1]. However, the establishment of a new catalytic route based on abundant transition metals should be economically more advantageous, given that high hydrogenation performance can be achieved under stable operation. Both liquid-phase batch mode [13,27,28] and vapor-phase continuous-flow [1,29] processes have been proposed in the earlier literature. Fixed-bed continuous-flow operation is more favorable in terms of ease of operation and lack of necessity for catalyst separation. As a compromise, higher temperatures are frequently adopted compared to those in the batch system. Under such reaction conditions, additional issues might arise, such as the coke deposition that has been observed over the Co/C-SiO$_2$ catalyst in the hydrogenation of nitrobenzene at 498 K [30]. Furthermore, HCl vapor can be generated by the hydrodechlorination reaction, and this by-product may represent an additional challenge to catalyst stability, particularly for non-precious-metal-based catalysts. Unfortunately, this aspect has not received enough attention from scientists. Thus, the lack of knowledge regarding the impact of HCl on catalyst stability might hinder the further development of a vapor-phase continuous-flow chloronitrobenzene hydrogenation process based on cheap metal catalysts.

Herein, we report a vapor-phase continuous-flow process for the chemoselective hydrogenation of 1,2-dichloro-4-nitrobenzene (DCNB) to the target product, 3,4-dichloroaniline (DCAni) over supported nickel catalysts at ambient pressure. Owing to the complex reaction network comprising hydrogenation and multiple hydrodechlorination paths (Figure 1), this reaction can serve as an ideal model reaction to assess both the chemoselectivity and stability of the catalysts. We found that the accumulation of substantial amounts of chlorine on the catalyst surface might block the access of the reactant molecules to the active metal sites, which was the main reason leading to fast catalyst deactivation. An effective strategy based on pre-coking the nickel catalyst under methane dry reforming ambient conditions was further developed to ease the site blocking caused by the HCl. As a result, the catalyst presented much improved stability performance in the selective hydrogenation reaction.

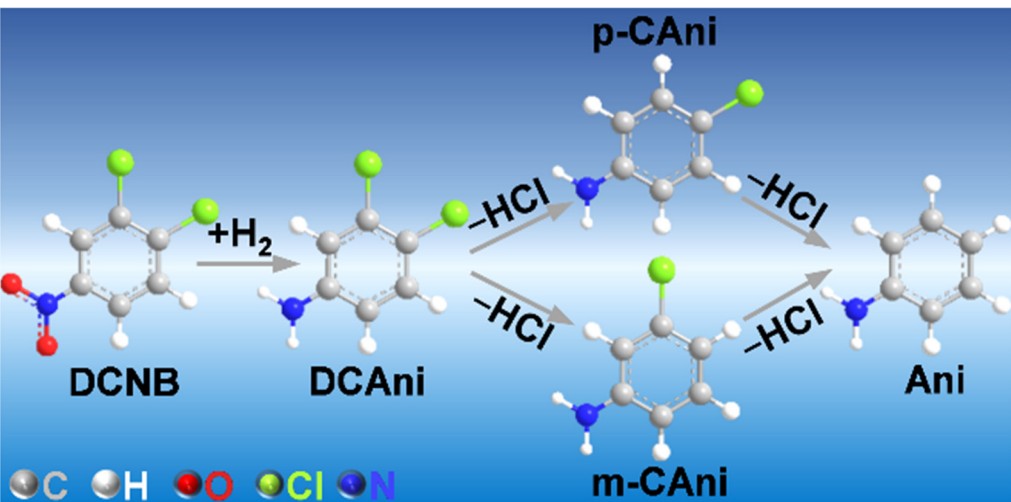

**Figure 1.** The reaction pathways for the hydrogenation of 1,2-dichloro-4-nitrobenzene (DCNB) to 3,4-dichloroaniline (DCAni). Other hydrodechlorinated by-products (m- and p-chloranilines and aniline, abbreviated as m- and p-CAni and Ani, respectively) can be formed by deep hydrogenation.

## 2. Results and Discussion

### 2.1. Synthesis and Characterization of Fresh Catalysts

The ZrO$_2$-supported nickel catalyst was prepared by the incipient impregnation method, using a nickel nitrate aqueous solution with self-prepared ZrO$_2$ synthesized by co-precipitation. Calcination of the dried sample at 400 °C yielded the expected precursor NiO/ZrO$_2$, as verified by the X-ray diffraction technique (XRD, Figure S1). Transmission electron microscopic (TEM, Figure S2) analysis revealed the relatively homogeneous morphology of the catalyst granules of 10–20 nm in size and also confirmed the different

crystallographic natures of the multiple components (NiO, and monoclinic (m) and tetragonal (m) ZrO$_2$). Reduction of the sample in a fixed-bed reactor in situ at 500 °C led to the formation of metallic nickel particles, reflected by the appearance of a small diffraction peak at 44° 2$\theta$ in XRD (Figure 2a). To introduce carbon onto the nickel particles, the reduced Ni/ZrO$_2$ catalyst was further exposed to a mixture of CH$_4$/CO$_2$ with a molar ratio of 1:1 at 750 °C for 30 min, and then cooled naturally to room temperature in a nitrogen stream. The derived sample was denoted as Ni/ZrO$_2$@C. The dry reforming reaction treatment induced an apparent alteration of the ZrO$_2$ support, i.e., the XRD reflection lines corresponding to the tetrahedral phase became more prominent than those of the monoclinic phase. Furthermore, the diffraction peaks associated with Ni$^0$ were sharpened after the treatment (Figure 2a). The surface species of the catalysts before and after the pre-coking treatment were probed using X-ray photoelectron spectroscopy (XPS, Figure 2b,c, and Table 1). The C 1s spectrum of Ni/ZrO$_2$ showed an asymmetric peak around 284.8 eV, accompanied by an envelope between 288 and 290 eV. The latter can be assigned to the C=O groups, which might be due to the formation of carbonates during the synthesis or post synthesis. Furthermore, the broad peak disappeared on Ni/ZrO$_2$@C. A possible reason is the decomposition of the carbonates during the high-temperature reforming reaction. The Ni 2$p_{3/2}$ spectra of the two catalysts were then compared. While both samples presented a major contribution around 852–859 eV accompanied by a satellite peak at 860–868 eV, surprisingly, the spectrum of Ni/ZrO$_2$@C showed a significant shift of about 1.5 eV to the high binding energy, compared with that of Ni/ZrO$_2$. This implies that the surface nickel species were likely to be more positively charged or that electron transfer might occur from Ni to the newly formed protective carbon materials.

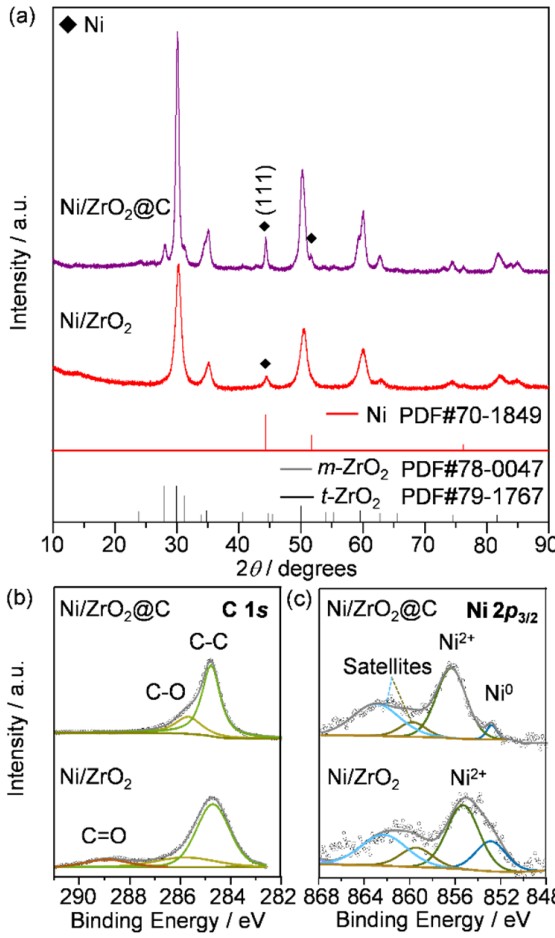

**Figure 2.** (**a**) XRD patterns, and (**b**) the C 1s and (**c**) Ni 2$p_{3/2}$ XPS spectra of Ni/ZrO$_2$ and Ni/ZrO$_2$@C.

**Table 1.** Characterization data of the fresh and used catalysts.

| Catalysts | $d_{Ni(111)}$ [a] /nm | $S_{BET}$ [b] /m$^2$ g$^{-1}$ | $V_p$ [c] /cm$^3$ g$^{-1}$ | C [d,e] /at.% | Ni [d,e] /at.% | Zr [d,e] /at.% | Cl [d,e] /at.% | O [d,e] /at.% |
|---|---|---|---|---|---|---|---|---|
| NiO/ZrO$_2$ | - | 74 | 0.25 | 22.24 | 2.65 | 18.08 | 0 | 57.03 |
| Ni/ZrO$_2$ | 8 | 94 | 0.20 | 25.37 (22.14) | 1.74 (7.45) | 19.38 (23.37) | 0 (0.05) | 53.51 (47.00) |
| Ni/ZrO$_2$-used | 12 | 98 | 0.25 | 30.70 (24.78) | 4.00 (5.71) | 12.28 (19.46) | 15.78 (3.33) | 37.24 (46.72) |
| Ni/ZrO$_2$@C | 19 | 41 | 0.23 | 55.45 (40.76) | 1.58 (6.02) | 10.61 (17.46) | 0 (0.01) | 32.36 (35.57) |
| Ni/ZrO$_2$@C-used | 23 | 60 | 0.25 | 67.10 (63.11) | 1.97 (3.58) | 5.36 (9.86) | 6.40 (0.96) | 19.16 (22.49) |

[a] The crystallite size of Ni in the catalysts estimated by the Scherrer equation using the Ni (111) facet. [b] Specific surface areas of the catalysts calculated from N$_2$ sorption. [c] The total pore volume of the catalysts determined from the amount of N$_2$ adsorbed at $p/p_0$ = 0.97. [d] The atomic concentration of the catalysts estimated by XPS. [e] The atomic concentration of the catalysts estimated by EDS, in parentheses.

The two catalysts were further examined by combined electron microscopic analysis (Figures 3 and 4). Scanning electron microscopic (SEM) analysis combined with electron dispersive spectroscopy (EDS) revealed a striking difference between the two catalysts. The surface of the coked sample became much rougher compared with that of Ni/ZrO$_2$.

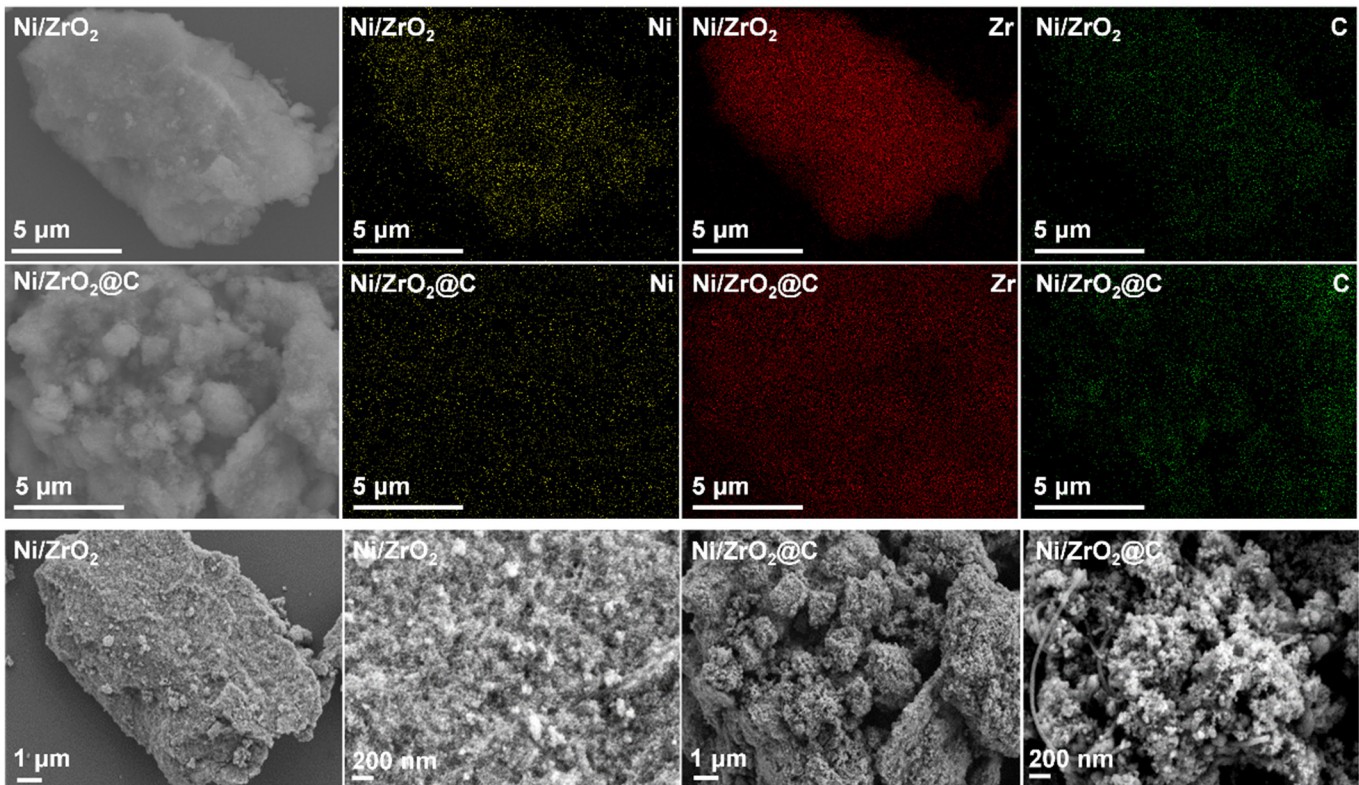

**Figure 3.** SEM images with elemental color mapping of Ni/ZrO$_2$ and Ni/ZrO$_2$@C.

Furthermore, the color mapping analysis showed that the carbon element was homogeneously distributed on Ni/ZrO$_2$, thus confirming that it indeed originated from the sample, which was consistent with the above XPS observation. On the other hand, the carbon distribution was not as homogeneous as Ni or Zr on Ni/ZrO$_2$@C. The C intensity was stronger on the rough surfaces, suggesting that these rough ensembles were likely the carbonaceous materials newly formed by the reforming reaction. Indeed, a close examination of the SEM images confirmed the fragmentation of the samples, as well as the presence of some tube-like carbon deposits. These clear morphology differences were further confirmed by TEM (Figure 4). Small nickel nanoparticles deposited on ZrO$_2$ were found on Ni/ZrO$_2$. In contrast, hollow carbon nanotubes were evidenced on Ni/ZrO$_2$@C. High-resolution TEM revealed that these nanotubes were composed of multi-walled graphite growing along the metallic nickel nanoparticles. This phenomenon is typically observed in the

dry reforming of methane over Ni-based catalysts, owing to the strong methane-splitting ability of the metal, and is a major cause of catalyst deactivation [31,32]. Since the nickel particles were covered by the protective carbon layers in this case, one can expect that the resulting catalysts might find applications in fields where the unique structure will bring additional benefits.

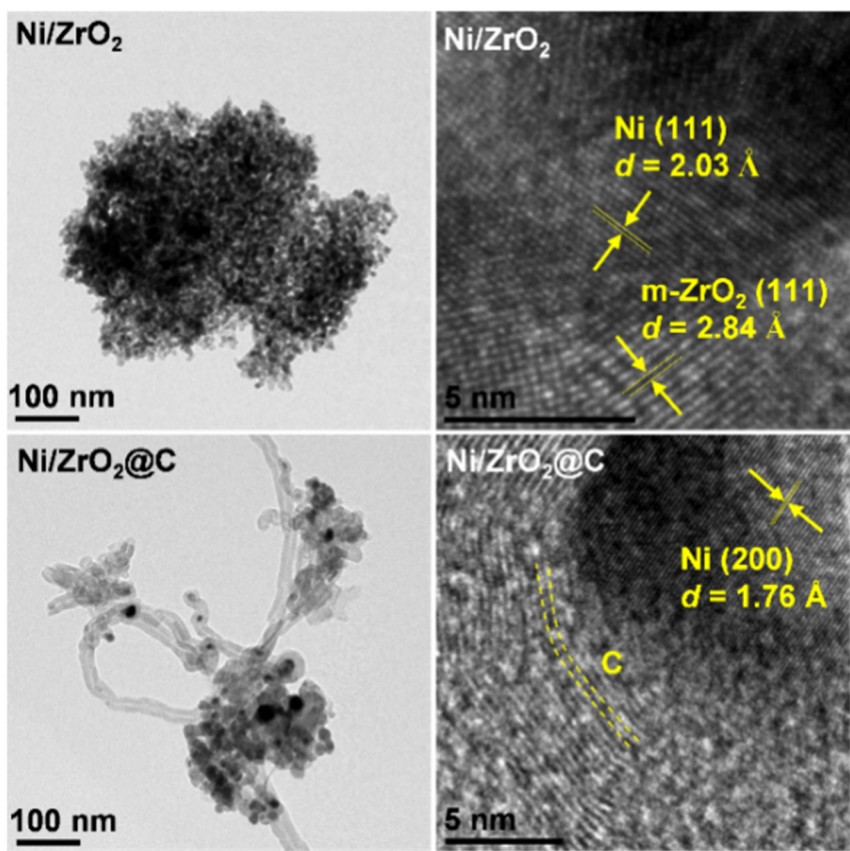

**Figure 4.** TEM images of Ni/ZrO$_2$ and Ni/ZrO$_2$@C.

### 2.2. Catalytic Performance in 1,2-Dichloro-4-nitrobenzene Hydrogenation

The gas-phase selective hydrogenation of 1,2-dichloro-4-nitrobenzene (DCNB) to 3,4-dichloroaniline (DCAni) was selected as a model reaction to assess the benefit of the carbon protective layer. We noted that although many different solid catalysts have been reported for the selective hydrogenation of different functionalized nitrobenzenes in the literature [1,2,5,11], there are no widely recognized benchmark systems. In particular, studies on the continuous-flow hydrogenation of DCNB were scarce, though there exists a very recent paper by Duan and co-workers [33]. For these reasons, pristine Ni/ZrO$_2$ was used as the reference to assess the effectiveness of the carbon deposits. Firstly, the catalytic activity was evaluated using temperature-ramping experiments from 50 to 300 °C over both catalysts. The pristine Ni/ZrO$_2$ was quite active, delivering ca. 50% conversion value at 50 °C and approaching full conversion at 150 °C (Figure 5a). The high activity was maintained up to 250 °C and then dropped at even higher temperatures. A similar trend was observed for the yield of the target product, DCAni. Detailed product distribution revealed that DCAni was the major product, accompanied by the other hydrodechlorinated by-products of m- and p-chloranilines and aniline (abbreviated as m- and p-CAni and Ani, respectively, as shown in Figure 5b). The best selectivity of DCAni was achieved between 125 and 175 °C, and elevated temperatures led to the formation of more hydrodechlorinated products. On the other hand, Ni/ZrO$_2$@C did not show apparent catalytic activity until the temperature was above 100 °C. Afterwards, the activity quickly built up and even surpassed that of the pristine catalyst. Interestingly, the activity was maintained even

up to 300 °C, thus apparently outperforming $Ni/ZrO_2$ at high temperatures. In addition, the yield of DCAni presented a similar trend as for the conversion of DCNB. Notably, the yield was always higher over $Ni/ZrO_2@C$ than $Ni/ZrO_2$ above 125 °C. These results suggested that the hydrogenation ability of $Ni/ZrO_2@C$ was weakened, compared to that of $Ni/ZrO_2$. The carbon protective layers on $Ni/ZrO_2@C$ are one of the suspected reasons for this. However, the particle size of the nickel might also play a role here, as the estimated crystallite sizes of Ni by XRD were 8 and 19 nm (Table 1), respectively, for $Ni/ZrO_2$ and $Ni/ZrO_2@C$. A striking feature of $Ni/ZrO_2@C$ is the high selectivity for DCAni, as was reflected by the product distribution. Unlike $Ni/ZrO_2$, $Ni/ZrO_2@C$ almost exclusively gave DCAni below 100 °C. The selectivity for DCAni was generally much higher at higher temperatures (except at 300 °C). Under these conditions, more by-product m-CAni was formed on $Ni/ZrO_2$, while more p-CAni was observed on $Ni/ZrO_2@C$. The high selectivity for the target product over $Ni/ZrO_2@C$ might be owing to the weakened hydrogenation ability restricting over-hydrogenation, and the manipulation of product selectivity between p- and m-CAni suggests an additional steric effect, probably caused by the carbonaceous deposits. The stability was then evaluated under even harsher conditions of 200 °C and four-fold liquid weight hourly space velocity (LWHSV) of 4 $h^{-1}$, to better compare the performance between the two catalysts (Figure 5c). $Ni/ZrO_2$ underwent a fast deactivation within 5 h and the activity dropped to 46%. In stark contrast, $Ni/ZrO_2@C$ was much more durable: the activity only dropped to 62%, even after 20 h. This demonstrated the significantly enhanced stability performance of $Ni/ZrO_2@C$ with the pre-deposited coke. In addition, the selectivity of DCAni (ca. 94–96%) was not much influenced during the course of the reaction over either catalyst.

### 2.3. Interpretation of the Deactivation Mechanisms

To explain the superior catalytic performance of $Ni/ZrO_2@C$ over $Ni/ZrO_2$ in the hydrogenation of DCNB, the catalysts were thoroughly characterized by different techniques after the stability tests. $N_2$ sorption revealed no significant alteration of the specific surface area ($S_{BET}$) for $Ni/ZrO_2$ (94 and 98 $m^2$ $g^{-1}$, respectively, before and after the test) but the $S_{BET}$ slightly increased from 41 to 60 $m^2$ $g^{-1}$ for $Ni/ZrO_2@C$ (Table 1). In addition, XRD showed that the diffraction intensity associated with $Ni^0$ increased for both the used catalysts (Figure 6a). Estimation of the crystallite size via the Scherrer equation (using the Ni(111) facet) indicated 12 and 23 nm, respectively, for $Ni/ZrO_2$ and $Ni/ZrO_2@C$ (Table 1). Despite the different durations of the tests (5 and 20 h), the growth of the particle size was more pronounced for $Ni/ZrO_2$ (150%) than for $Ni/ZrO_2@C$ (121%). This suggests that the agglomeration of the nickel nanoparticles was greatly retarded by the covered carbon shells. Furthermore, some small diffraction peaks related to carbon were also visualized on the used $Ni/ZrO_2@C$ via XRD. Since coke deposition was disclosed in the hydrogenation of nitrobenzene in a previous study [30], it was necessary to verify whether this was one of the reasons for the catalyst deactivation. The used catalysts were then checked by SEM and TEM (Figure 6b). Both techniques revealed that the catalyst morphologies were essentially the same before and after the catalytic tests. HRTEM confirmed the clean edge of metallic nickel on $Ni/ZrO_2$ and the multiple-carbon-shell-covered $Ni^0$ on $Ni/ZrO_2@C$. These results were corroborated by SEM. No obvious carbonaceous deposits were found on the used $Ni/ZrO_2$. However, the proportion of carbon nanotubes seemed to be increased on the used $Ni/ZrO_2@C$, based on SEM.

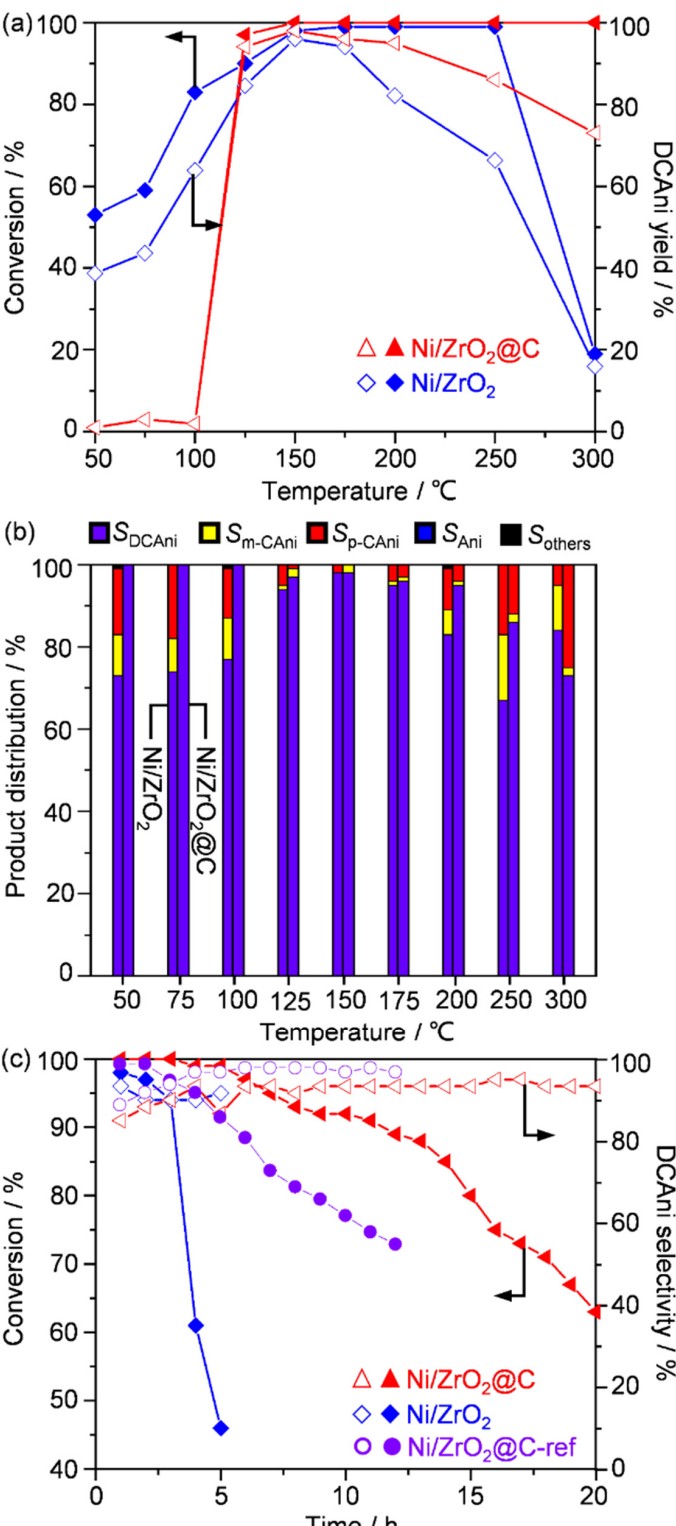

**Figure 5.** Catalytic performance in the continuous-flow hydrogenation of DCNB over different nickel catalysts. (**a**) The conversion and DCAni yield, and (**b**) product distribution as a function of the bed temperature. (**c**) Stability performance. Reaction conditions: $P$ = 1 bar, (**a**,**b**) $F_{H2}$ = 10 mL min$^{-1}$ and $F_{liquid}$ = 1 mL min$^{-1}$, (**c**) $F_{H2}$ = 40 mL h$^{-1}$ and $F_{liquid}$ = 4 mL h$^{-1}$, and $T$ = 200 °C. The same weight of nickel was ensured in the hydrogenation by mixing the pristine (500 mg) and pre-coked catalysts with quartz sand of similar particle size, and the total weight was fixed at 1.0 g. The liquid was 5 wt.% DCNB in toluene.

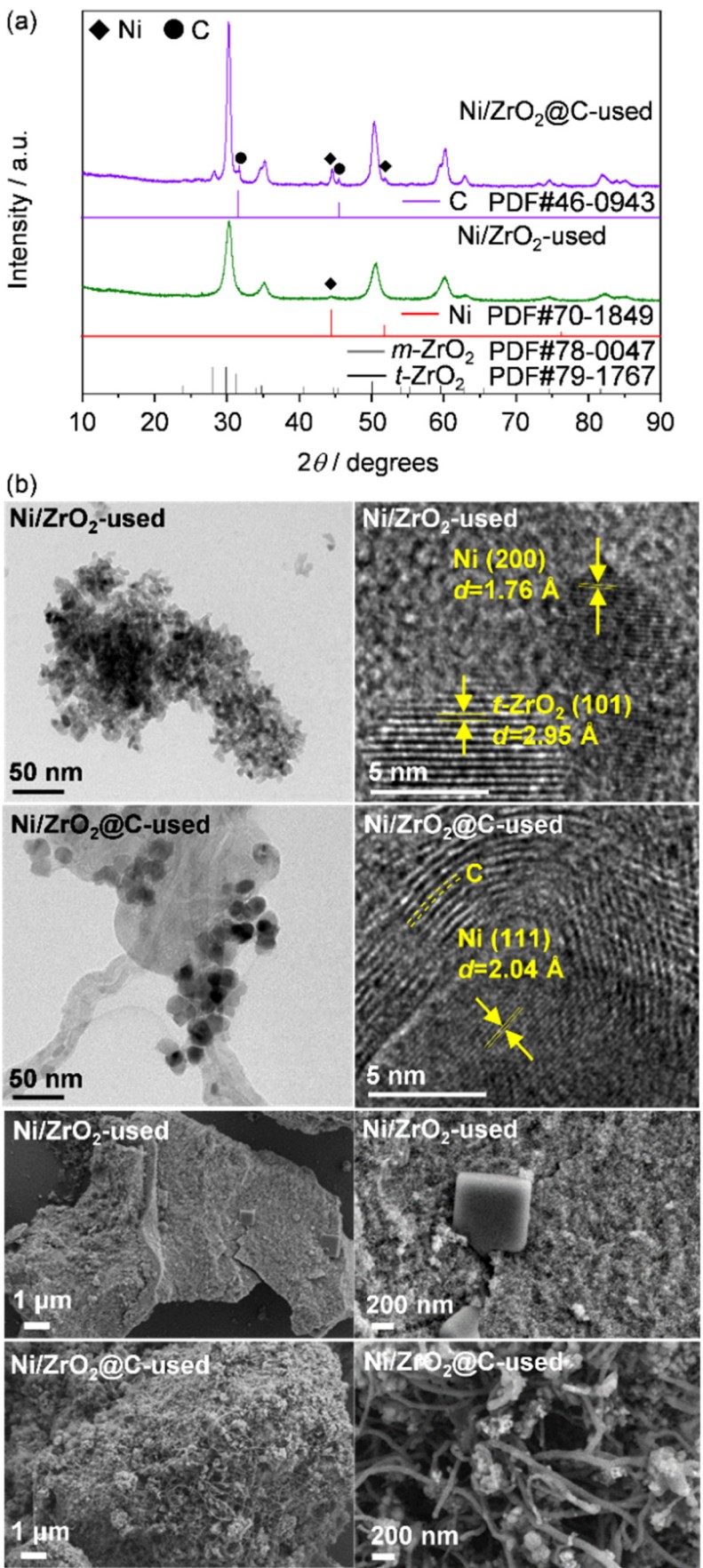

**Figure 6.** (**a**) XRD, (**b**) TEM and SEM images of Ni/ZrO$_2$ and Ni/ZrO$_2$@C after the stability tests.

To further corroborate the electron microscopic results, Raman spectroscopic analyses were performed (Figure 7a). Distinct features at 1340 and 1580 cm$^{-1}$, which can be assigned to the *D* and *G* bands, respectively, of carbon materials, were present for the fresh and used Ni/ZrO$_2$@C. The intensities of these bands were strengthened, compared to those of the fresh catalyst, and the relative band intensity also showed a difference ($I_D$:$I_G$ = 0.94 and 0.84, respectively, before and after the test). In contrast, no clear Raman peaks could be observed on the fresh and used Ni/ZrO$_2$ catalysts. Therefore, carbon deposition probably occurred on Ni/ZrO$_2$@C but not on Ni/ZrO$_2$, agreeing well with the above electron microscopic analyses (Figure 6b). Thermogravimetric analyses (TG) were conducted on the catalysts to quantify the organic and inorganic species (Figure 7b). The used Ni/ZrO$_2$ catalyst showed three-stage weight loss. The first stage occurred below 150 °C with a weight loss of 6.5%, which can be assigned to the removal of moisture. The second and third stages occurred below 484 and 980 °C with weight losses of 6.2 and 3.9%, respectively, which might be related to surface chlorinated compounds, as is discussed later. The pre-coked Ni/ZrO$_2$@C presented two-stage weight loss: the low-temperature weight loss of 2.0% was related to moisture and the high-temperature (400–600 °C) loss of 5.7% was due to the removal of pre-deposited carbon nanotubes. The weight loss behavior of the used Ni/ZrO$_2$@C was much more complicated. It consisted of a low-temperature weight loss of 1.2% below 300 °C and three consecutive weight losses of 4.3, 7.6, and 2.5% below 480, 612, and 980 °C, respectively. The latter three weight losses might be due to the combined removal of carbon deposits and chlorinated compounds. The final product of this sample after TG analysis was checked by XRD (Figure S3), and the presence of NiO and ZrO$_2$ was confirmed. By excluding the weight loss caused by moisture (the first-stage weight loss) and the pre-deposited coke (5.7%), the calculated weight losses caused by chlorinated compounds and the newly deposited coke were 10.1 and 8.7%, respectively, for the used Ni/ZrO$_2$ and Ni/ZrO$_2$@C catalysts. Considering the much-shortened reaction time over Ni/ZrO$_2$, the result hinted that the accumulation of chlorine was much faster on this sample.

To support the above conclusion, the content of chlorine and carbon on the catalysts was semi-quantified by EDS (Figure 7c) and XPS (Table 1). Although the data were not fully consistent for both techniques owing to the different penetration depths (XPS is more surface sensitive and EDS is more related to the bulk), similar trends can still be generated from Table 1. By comparing the content of C and Cl for the catalysts before and after the tests, one can see that the carbon content only increased moderately for the used Ni/ZrO$_2$ but built up prominently for the used Ni/ZrO$_2$@C. Therefore, this finding supported the conclusion that coke deposition might be absent on Ni/ZrO$_2$ but occurred on Ni/ZrO$_2$@C, which is consistent with the Raman and SEM results. In contrast, the Cl content was almost three times higher on the used Ni/ZrO$_2$ than on the used Ni/ZrO$_2$@C, confirming the much more pronounced uptake of chlorine, and thus supporting the TG analysis.

Taking account of the above characterization results, we speculate that the deactivation mechanisms were different over Ni/ZrO$_2$ and Ni/ZrO$_2$@C. The accumulation of chlorine was the major cause over Ni/ZrO$_2$, and additional accompanying coke deposition over Ni/ZrO$_2$@C. Considering that the deactivation was much faster over Ni/ZrO$_2$, the chlorine accumulation should play a key role in the deactivation process. To verify this point, we prepared an additional pre-coked catalyst reference (Ni/ZrO$_2$@C-ref) with more deposited coke, by extending the treatment time to 2 h (instead of half an hour) during the methane dry reforming reaction. This reference catalyst, despite containing even more coke (20.7 wt.% according to TG, as shown in Figure S4) than the used Ni/ZrO$_2$@C, still delivered full conversion of DCNB at the beginning, then gradually deactivated. Although it showed poorer stability performance compared to Ni/ZrO$_2$@C, it was still better than Ni/ZrO$_2$. Thus, these results convincingly suggest that the chlorine-induced catalyst deactivation mechanism prevailed. We expect that thermal activation in protective gases or hydrogen treatment might be effective in removing the chlorine on the catalyst, at least partly restoring the activity. However, repetitive catalyst regeneration can also be tedious and energy consuming. Nanostructuring of the supported metal catalysts with new

compositions and structures to restrict chlorine uptake might lead to a more promising technology for continuous-flow hydrogenation of these specially functionalized substrates.

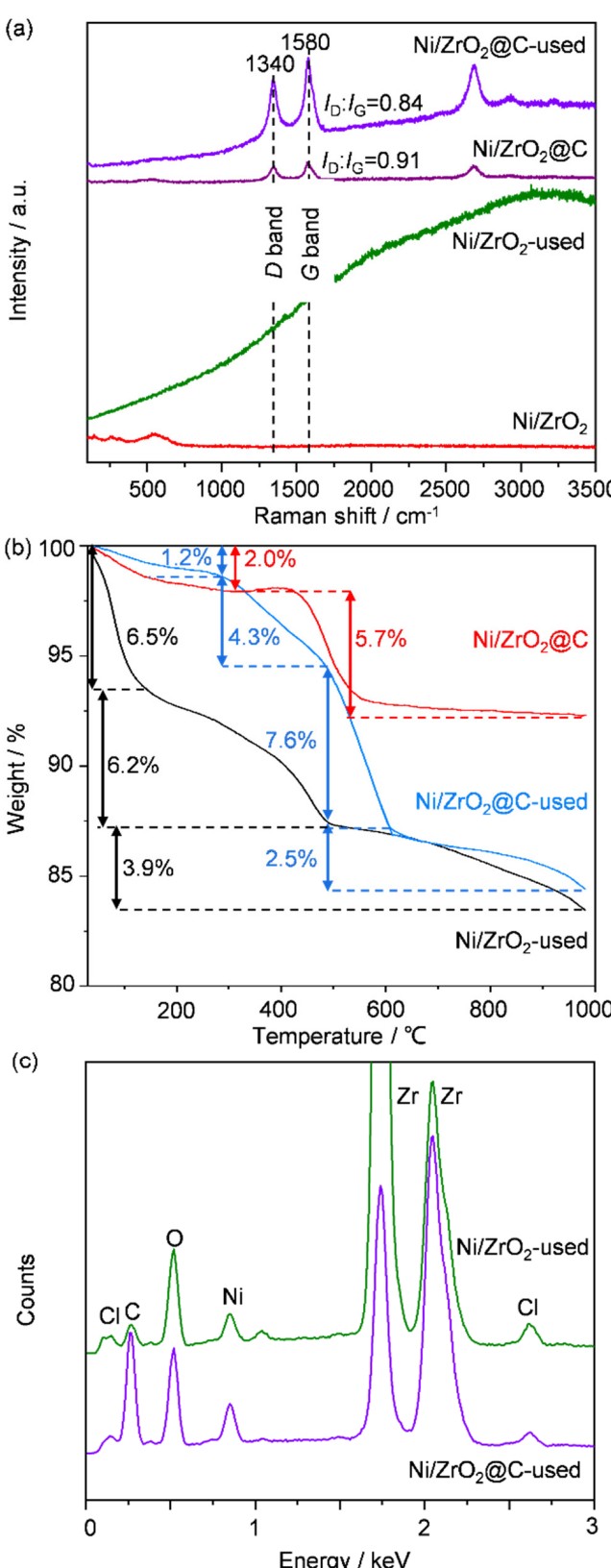

**Figure 7.** (**a**) Raman, (**b**) TG, and (**c**) EDS analyses of Ni/ZrO$_2$ and Ni/ZrO$_2$@C after the stability tests.

## 3. Materials and Methods

### 3.1. Catalyst Preparation

The $ZrO_2$ support was prepared by a co-precipitation method. Typically, an aqueous metal nitrate ($Zr(NO_3)_4 \cdot 5H_2O$, AR, Macklin, Shanghai, China) and another aqueous solution of NaOH (2 mol $L^{-1}$, $\geq$96.0%, Shanghai, China) were prepared. Both solutions were co-added at a speed of 2 mL $min^{-1}$ into a rounded flask, kept at 30 °C, with vigorous stirring. The final pH of the solution was kept at 9. The solid precipitate was then aged at 90 °C for another 4 h, washed extensively with deionized water, filtered off, dried at 110 °C for 16 h, and finally calcined at 450 °C for 4 h to yield $ZrO_2$ oxides. The supported nickel catalyst was prepared by an incipient wetness impregnation method using the above derived oxides as the carriers. To a calculated weight of the carriers, an aqueous solution of $Ni(NO_3)_2 \cdot 6H_2O$ ($\geq$98.0%, Shanghai, China) was added with mild mixing. The solids were then dried at 110 °C for 16 h and calcined at 400 °C for 4 h to yield the supported catalyst with a nominal Ni loading of 10 wt.%.

### 3.2. Pre-Coking Strategy for Supported Nickel Catalysts in Dry Reforming of Methane

To introduce carbon onto the nickel particles, the reduced $Ni/ZrO_2$ catalyst was further exposed to a mixture of $CH_4/CO_2$ with a molar ratio of 1:1 at 750 °C for 30 min. Firstly, the $NiO/ZrO_2$ catalyst (20–40 mesh, 500 mg) was placed between two quartz wool plugs in the center of the quartz tube reactor. Before the pre-coking, the catalyst was reduced at 500 °C for 3 h in $H_2$ (30 $cm^3$ $min^{-1}$). Secondly, it was heated to 750 °C in a nitrogen stream (30 $cm^3$ $min^{-1}$). Thirdly, a concentrated feed with a molar $CH_4$:$CO_2$ ratio of 1:1 was admitted into the reactor at a gas hourly space velocity (GHSV) of 72,000 mL $g^{-1}$ $h^{-1}$ for 30 min and then cooled naturally to room temperature in a nitrogen stream. The derived sample was denoted as $Ni/ZrO_2$@C. An additional reference catalyst, $Ni/ZrO_2$@C-ref, was prepared following similar procedures to $Ni/ZrO_2$@C but a treatment time of 2 h was applied in the dry reforming reaction to deposit more coke.

### 3.3. Catalyst Characterization

Nitrogen sorption was measured at 77 K on a Quantachrome NT3LX-2 instrument (USA) after degassing the samples at 573 K for 3 h. The specific surface area was calculated using the Brunauer–Emmett–Teller (BET) method. X-ray diffraction (XRD) patterns were obtained using an X'Pert3 Panalytical X-ray diffractometer (PANalytical, Almelo, The Netherlands) using Cu K$\alpha$ radiation with a scanning angle ($2\theta$) range of 10–90° at a speed of 2 °C $min^{-1}$. The tube voltage and the current were 40 kV and 40 mA, respectively. The samples were tightly pressed onto a glass holder for the analysis. Thermogravimetric analyses (TG) were carried out on a NETZSCH STA 449 F3 (Selb, Germany) system with a heating rate of 10 °C $min^{-1}$ (from room temperature to 1000 °C) under flowing air. The samples were placed in a crucible made of $\alpha$-$Al_2O_3$ for the analysis. Raman spectra were acquired on a confocal laser micro-Raman spectrometer (HORIBA, LabRAM HR Evolution, Longjumeau, France) under the following conditions: a wavelength of 532 nm, a transit of 10 s per sample, and spectral resolution of 1 $cm^{-1}$. The sample powders were dispersed in ethanol by ultrasonication and then the specimen was obtained by dropping a droplet suspension onto a carbon film supported on a copper grid for TEM analysis. Scanning electron microscopy (SEM) was performed on a Hitachi S-5500 scanning electron microscope (Japan) operating at an acceleration voltage of 30 kV. X-ray photoelectron spectroscopic (XPS) analyses of the catalysts were carried out on a Thermo ESCALAB 250Xi spectrometer (Shanghai, China) using a 15 kV Al K$\alpha$ X-ray source as a radiation source. The binding energy was calibrated using the C 1s peak (284.8 eV) as the reference.

### 3.4. Catalyst Evaluation in Chloronitrobenzene Hydrogenation

The selective hydrogenation of 1,2-dichloro-4-nitrobenzene (DCNB) and 3,4-dichloroaniline (DCAni) was carried out in a fixed-bed down-flow reactor at atmospheric pressure. The catalysts were diluted with quartz sand of the same particle sizes. After being mixed

homogeneously, they were placed between two quartz wool plugs in the center of the quartz tube reactor. The catalytic performance was evaluated by temperature ramping between 50 and 300 °C. A concentrated feed of 5 wt.% DCNB in toluene was admitted into the reactor at a liquid weight hourly space velocity (LWHSV) of 1 h$^{-1}$. The stability tests were performed at 200 °C at a LWHSV of 4 h$^{-1}$. The reaction products were analyzed using a gas chromatograph (Agilent 7890B, Shanghai, China) fitted with an FID detector using a HP-5 column.

## 4. Conclusions

We demonstrated a new pre-coking strategy to manipulate the catalytic performance of supported nickel catalysts for the selective continuous flow hydrogenation of 1,2-dichloro-4-nitrobenzene to 3,4-dichloroaniline. By introducing controlled tube-like coke along the nickel particles through a methane dry reforming reaction, we demonstrated that the pre-coked catalyst showed weakened hydrogenating ability that could be offset by increasing the reaction temperature. In addition, pre-coking significantly improved the thermal stability of the nickel particles compared to the pristine analog, probably due to the physical restraining effect of the coke. Different deactivation mechanisms were disclosed for the pristine and pre-coked catalysts. While significant chlorine uptake occurred on the pristine Ni/ZrO$_2$, this effect was much milder on Ni/ZrO$_2$@C, over which additional coke deposition was found in the hydrogenation. Compared to the coke deposition, the accumulation of chlorine was found to be more crucial in accelerating catalyst deactivation. The enhanced stability performance of Ni/ZrO$_2$@C in the hydrogenation can be thus ascribed to the retarded uptake of chlorine from the hydrodechlorination reactions. Therefore, these fundamental findings are expected to stimulate new interest in specific hydrogenation applications where halides are involved.

**Supplementary Materials:** The following are available online at https://www.mdpi.com/article/10.3390/catal11101156/s1, Figure S1: the XRD patterns of as-prepared ZrO2 and NiO/ZrO2, Figure S2: the TEM image of NiO/ZrO2, revealing the presence of nanoparticles of NiO on the ZrO2 carrier, Figure S3: the TG profiles of Ni/ZrO2@C and Ni/ZrO2@C-ref. The numbers indicate the weight losses of the carbon deposits.

**Author Contributions:** Conceptualization, X.M. and Y.D.; methodology, X.M.; investigation, P.W. and S.W.; writing—original draft preparation, P.W.; writing—review and editing, R.L., X.M., and Y.D. All authors have read and agreed to the published version of the manuscript.

**Funding:** This research received no external funding.

**Conflicts of Interest:** The authors declare no conflict of interest.

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
