# Peer review of "Pre-Coking Strategy Strengthening Stability Performance of Supported Nickel Catalysts in Chloronitrobenzene Hydrogenation"

_catalysts, doi:10.3390/catal11101156_

Round 1

Reviewer 1 Report

The authors report a vapor-phase continuous-flow process for the chemoselective hydrogenation of 1,2-dichloro-4-nitrobenzene (DCNB) into the targeted product 3,4-di-chloroaniline (DCAni) over the support nickel catalysts at ambience pressure. 

The manuscript is very interesting and well organized. I would suggest some minor changes before accepting it for publishing:

Please change X-Ray diffraction into Powder X-Ray diffraction through all the text. XRD usually refers to the monocrystalline samples, which is not the case here. Which kind of holder was used for PXRD experiments? Please insert it into the experimental section.

TG analysis please add which kind of crucible was used for the analysis. Further could you corellate the experimental weight loss values with the theoretical one, especially for the first step that should correspond to the moisture? What was the final product after the TG anaylsis and how was it confirmed?

Figures 5 a and c: please insert the legend at the graphs since its hard to follow the curves. 

Reviewer 2 Report

This paper describes a pre-coking strategy for reducing the deactivation of Ni/Zirconia catalyst in chlornitrobenzene hydrogenation. While adequate studies have been carried out to support the authors' hypotheses, the one thing lacking from this paper is any kind of benchmarking. How effective is this catalyst, both before and after pre-coking, in comparison to others used in literature for a similar application? How do the TON/TOFs compare? How does pre-coking affect product selectivities? What are some of the other strategies used to regenerate similar catalysts after poisoning by halides (or prevent such poisoning in the first place)? How effective is this protocol in comparison to existing literature protocols?

I'd like to see these concerns addressed before the manuscript is accepted for publication.

Round 2

Reviewer 2 Report

accept as-is.